# Combination of Optimized Variational Mode Decomposition and Deep Transfer Learning: A Better Fault Diagnosis Approach for Diesel Engines

**Huajun Bai [1], Xianbiao Zhan [1], Hao Yan [1], Liang Wen [1,2] and Xisheng Jia [1,\***]

[1] Shijiazhuang Campus, Army Engineering University of PLA, Shijiazhuang 050003, China; baihuajun001@gmail.com (H.B.); zhanxianbiao001@gmail.com (X.Z.); yanhao202201@gmail.com (H.Y.); wenliang202201@gmail.com (L.W.)
[2] Hebei Key Laboratory of Condition Monitoring and Assessment of Mechanical Equipment, Shijiazhuang 050003, China
[\*] Correspondence: jiaxisheng001@gmail.com; Tel.: +86-0311-87994788

**Abstract:** Extracting features manually and employing preeminent knowledge is overly utilized in methods to conduct fault diagnosis. A diagnosis approach utilizing intelligent methods of the optimized variational mode decomposition and deep transfer learning is proposed in this manuscript to deal with fault diagnosis. Firstly, the variational mode decomposition is optimized by $K$ values of the dispersion entropy to realize an adaptive decomposition and reduce the noise of the signal. Secondly, an image with two dimensions is generated by a vibration signal with one dimension utilizing a short-time Fourier transform, after conducting noise reduction. Then, the ResNet18 network model is used to pre-train the model. Finally, the model transfer method is used to detect faults of a diesel engine. The results show that the proposed method outperforms the deep learning methods available in the literature. Besides, better noise reduction ability and higher diagnostic accuracy are attained.

**Keywords:** variational mode decomposition; short time Fourier transform; transfer learning; deep residual network; diesel engine troubleshooting

## 1. Introduction

Diesel engines are generally utilized in the construction of machinery, automobiles, vessels, and in other production areas. Due to their complex design structure and their long-term operation in harsh environments, various failures will inevitably occur, which lead to prolonged equipment downtime and increased maintenance costs. So, dealing with these issues has become a hot research direction, contemporarily studied by many scholars. Moreover, various intelligent data-driven approaches to coping with fault diagnosis of diesel engine failures have been suggested and satisfactory research outcomes have been achieved [1–3].

Fault diagnosis approaches with machine learning methodologies substantially include three fault facets: extraction of features, reduction of feature dimensionality, and pattern recognition of features. For these purposes, the time-frequency analysis of the vibration signal is studied, the commonly implemented algorithms of which are called Hilbert Huang Transform (HHT), Wavelet Transform (WT), and Short Time Fourier Transform (STFT) [4–6]. The implementations of these algorithms aim at extracting the feature parameters of both time and frequency domains [7,8]. When feature dimensionality reduction is under consideration, principal component analysis (PCA), kernel principal component analysis (k-PCA), and autoencoder are generally implemented [9–11]. Similarly, when pattern recognition is under consideration, support vector machine (SVM), random forest (RF), and k-nearest neighbor (k-NN) methods are commonly utilized [12–14]. Thus, these

feature extraction and dimensionality reduction techniques are combined to detect pattern recognition of various failures. Moreover, various diagnostic methods to detect faults in diesel engines have been suggested [15–17].

The conventional diagnosis methods utilizing intelligent approaches constitute a small set of machine learning methods with higher diagnosis accuracies. However, they are also accompanied by restrictions, as follows:

(1) Due to a large quantity of both noise and interference available in sampled vibration signals, detection of weak fault signals is generally difficult. To cope with weak fault signals, more advanced signal preprocessing techniques must be employed [18].

(2) If the feature parameters are set improperly, or rely on largely preeminent knowledge of experts [19,20], the accuracy of fault diagnosis is affected.

As artificial intelligence technology rapidly advances, deep learning has gradually turned into an efficient method and surmounts the deficiencies of the conventional fault diagnosis approaches. So, extracting useful fault features directly from raw data is a key advantage. Hence, deep belief network (DBN), convolutional neural network (CNN), and long short-term memory network (LSMN) are commonly implemented to diagnose faults related to mechanical applications [21,22]. The DBN was implemented by Xu et al. [23] to diagnose the air path fault of turbofan engines with higher classification accuracy. The parameters of the CNN were optimized by Zhou et al. [24], by utilizing the sorting method of input measurement parameters, which was applied to detect the fault of the gas circuit of an engine with a relatively ideal diagnosis impact. Han et al. [25] constructed a data-driven fault prediction model using an LSTM network and applied it to marine diesel engines, and obtained better fault prediction results.

Theproblems expressed below still exist and need to be dealt with:

(1) Since diesel engines work in complex environments for a long time, weak fault features are masked by stronger noise and interference signals, which greatly increases the difficulty of direct fault diagnosis using one-dimensional vibration signals.

(2) The parameters in network training increase with the increment of the number of hidden layers. When a multi-layer deep network model is trained, both preparing labeled samples in large quantities and the requirements of computational power and time need to be taken into account. However, collecting a large amount of data with faulty tags is almost impossible when the equipment basically runs in a steady state, in which few failures could occur.

When training a multi-layer deep network model from scratch, there is not only a need to prepare a large number of labeled samples, but also the training consumes a lot of computing power and time. Then, in practical engineering applications, the equipment is running in a normal condition, and there exist few failures, which makes it impossible to obtain a large number of data samples with faulty tags.

(3) The hyperparameter optimization and selection of the deep learning network model consumes a lot of time in training the network model and, thus, this will directly affect its performance.

Due to the issues mentioned above, a method called transfer learning (TL) was suggested when mechanical fault diagnosis is under consideration. Xu et al. [26] suggested an approach employing migration component analysis to determine fault diagnosis when various working conditions were taken into account. Zhao et al. [27] realized cross-domain aero-engine fault diagnosis by employing extreme learning machines, using the TL method. Xiong et al. [28] suggested a methodology utilizing stacked autoencoders and feature transfer to diagnose diesel engine faults. Both training and optimizing the deep learning network model are essential when the available TL research is under consideration, which restricts its application in engineering implementations.

To resolve the issues presented above, this manuscript suggests a methodology to diagnose faults utilizing both optimized variational mode decomposition (VMD) and deep transfer learning (DTL), concurrently. Firstly, the VMD method is optimized by using the $K$ value of the dispersion entropy to conduct the noise reduction in the original vibration.

Then, the noise-reduced vibration signal is converted into a frequency map with two dimensions by the STFT method. To lower both training time and computational complexity of the deep learning network model, a TL method, based on the ResNet18 network model, is suggested, which could effectively extract useful features from the frequency map represented by two dimensions, and quickly achieve accurate fault classification. Finally, experiments conducted present both better-extracted features and higher diagnostic accuracies.

The key contributions of the manuscript are expressed as follows:

(1) The optimized VMD can not only withhold weak fault signals, but also better excludes both noisy and interfering signals in the original vibration signal, which results in successful noise reduction.

(2) By employing the ResNet18 network model, trained on the ImageNet dataset as the transfer object, which provides a fine-tuning of the network structure, the training time of the network could be effectively lowered, and the accuracy of identifying faults could be improved.

(3) This research utilizes both optimized VMD and DTL concurrently to extract and classify fault features of diesel engines in strong noise environments. The proposed method exhibited greatly improved performance and a practically usable form to diagnose faults in applications in experiments.

The remaining sections of the manuscript are constructed as follows: A method dealing with noise reduction utilizing the optimized VMD is described in Section 2. Section 3 details the basic theory of time-frequency images and DTL models. The fault diagnosis process of a diesel engine, utilizing both optimized VMD and DTL, is described in Section 4. Section 5 analyzes and validates the method suggested in this manuscript employing pre-set experiments. The conclusion is provided in Section 6. Abbreviations provides the commonly used symbols in this manuscript.

## 2. A Noise Reduction Approach Utilizing the Optimal VMD

The VMD, Variational Mode Decomposition, as a novel non-stationary and non-linear signal processing methodology, was proposed in [29]. By resolving the constrained variational problem to replace the previous empirical mode decomposition (EMD), ensemble empirical mode decomposition (EEMD), and local mean decomposition (LMD) were suggested [30]. Hence, the modal aliasing and end effect problems of the conventional decomposition method were improved. However, the VMD itself also has certain limitations. The determination of its decomposition level K needs to be set manually, which has a certain degree of subjectivity and randomness that will have an impact on the modal decomposition process. Therefore, this paper adopts a method of optimizing the VMD based on the $K$ value of dispersion entropy to realize the adaptive decomposition and reconstruction of noise reduction of the signal.

### 2.1. The VMD

The function of the VMD is to establish and resolve the variational problem, and decompose the available signal $f$ into $K$ Intrinsic Mode Function (IMF) components, $u_k(t)$. Both epicenter frequency and bandwidth are continuously revised by running the process iteratively, since the summation of each component is assumed to be equal to the input signal. Then, the IMF component minimizing the summation of the bandwidths related to the IMF is obtained, as follows:

(1) By running the Hilbert transform, the $u_k(t)$ ($k = 1, 2, \cdots, K$) analytic signal unilateral spectrum of each IMF component is obtained, namely,

$$[\delta(t) + (j/\pi t)] \times u_k(t) \tag{1}$$

then, the position of the center frequency of each eigenmode to the corresponding baseband is expressed by

$$\{[\delta(t) + (j/\pi t)] \times u_k(t)\}e^{-j\omega_k t} \tag{2}$$

where the impulse function is denoted by $\delta(t)$, the sign "$*$" represents the convolution calculation, and $\omega_k$ represents the center frequency of each IMF component;

(2) The $L^2$ norm of the gradient of the demodulated signal is computed, and the bandwidth of each IMF component is estimated. Then, the constrained variational model expression is expressed by

$$
\left\{
\begin{array}{c}
\min\limits_{\{u_k, \omega_k\}} \left\{ \sum\limits_{k=1}^{K} \left\| \partial_t \left[ \left( \delta(t) + \frac{j}{\pi t} \right) \times u_k(t) \right] e^{-j\omega_k t} \right\|_2^2 \right\} \\
s.t \sum\limits_{k=1}^{K} u_k = f(t)
\end{array}
\right\}
\tag{3}
$$

where $n \, \|\,\|_2$ denotes the $L^2$ norm operation; $u_k = \{u_1, u_2, \cdots, u_k\}$ is the $k$ IMF component, and $f(t)$ represents the original time-domain signal; $\omega_k = \{\omega_1, \omega_2, \cdots, \omega_k\}$ represents the central frequency of each component

(3) Both $\alpha$ and $\lambda$ are used in the solution of the variational problem, which is called the quadratic penalty factor and the Lagrange multiplication operator. Then, the constrained variational problem is transformed into an unconstrained form, in which the Lagrange function in the augmented form is expressed by

$$
\begin{aligned}
L(\{u_k\}, \{\omega_k\}, \lambda) = \quad & \alpha \sum_k \left\| \partial_t \left[ \left( \delta(t) + \frac{j}{\pi t} \right) \times u_k(t) \right] e^{-j w_k t} \right\|_2^2 \\
& + \left\| f(t) - \sum_{k=1}^{K} u_k(t) \right\|_2^2 \\
& + \left( \lambda(t), f(t) - \sum_{k=1}^{K} u_k(t) \right)
\end{aligned}
\tag{4}
$$

where $\alpha$ represents the quadratic penalty factor, which is usually selected as a positive large number to enhance the accuracy of the reconstructed signal; $\lambda$ is called the Lagrange multiplication operator, and $\lambda(t)$ guarantees strict constraints. ADMM, the Alternate Direction Method of Multipliers, is employed to recompute the values $u_k$, $\omega_k$ and $\lambda$ in each component. Then, the saddle point of the augmented Lagrange function is calculated, namely, the constrained variational model is optimized to realize modal decomposition.

### 2.2. Dispersion Entropy

The principle of the VMD signal decomposition was introduced in the previous section. However, determining the VMD decomposition layers, $K$, needs to be set manually. So, $K$ would directly have an impact on the decomposition effect of the signal when a complex signal is processed. When $K$ is set to large numbers, over-decomposition could occur, thus redundant components would be obtained. On the other hand, when $K$ is assigned to small numbers, under-decomposition could happen and, thus, the useful signal could not be effectively separated. In this paper, it is proposed that $K$ is optimized by employing scatter entropy to find and conduct the decomposition of the signal adaptively.

DE, Dispersion Entropy, is a novel method suggested by Rostaghi and Azami in 2016 to measure the complexity of time series [31]. The fact that conventional permutation entropy does not consider the magnitude of amplitude was remedied and, thus, better stability and faster calculation were mentioned as advantages. The steps are presented by:

(1) The function of the normal distribution is selected as the nonlinear normalization function. The sequence $x = \{x_1, x_2, \ldots, x_N\}$ is normalized with the mean and standard deviation of $x$ as parameters. Then $y = \{y_1, y_2, \ldots, y_N\}$ is attained, where $N$ represents the sequence length with $y \in (0,1)$.

(2) Map $y$ to integers in the range [1,$c$] through a linear algorithm to obtain the sequence by

$$z_j^c = \text{int}(cy_j + 0.5) \tag{5}$$

where $c$ and int represent the category numbers and rounding, respectively.

(3) Compute both the embedded vector and the scatter pattern $w_{v_0 v_1 \ldots v_{m-1}}$ ($v$ = 1, 2, $\ldots$ ,$c$), and compute the probability $P$ for all scattered patterns defined by

$$P(w_{v_0 v_1 \ldots v_{m-1}}) = \frac{num(w_{v_0 v_1 \ldots v_{m-1}})}{N - (m-1)d} \tag{6}$$

where $z_i^c = v_0$, $z_{i+d}^c = v_1$, $\ldots$ , $z_{i+(m-1)d}^c = v_{m-1}$; $num(w_{v_0 v_1 \ldots v_{m-1}})$ are $z_i^{m,c}$ the number of mappings to scatter patterns, $m$ and $d$ represent the embedding dimension and the time delay, respectively.

(4) The original sequence *DE* is calculated by utilizing the definition of information entropy as follows:

$$DE(x, m, c, d) = -\sum_{w=1}^{c^m} P(w_{v_0 v_1 \ldots v_{m-1}}) \ln(P(w_{v_0 v_1 \ldots v_{m-1}}))) \tag{7}$$

According to the calculation method of the *DE*, the dispersion entropy, having maximum value when the probability of all dispersion modes is equal, is found. The larger the value of the dispersion entropy, the greater the complexity of the time series would be. Thus, the manuscript employs the dispersion entropy to optimize *K* of the VMD decomposition level. The turning point of the dispersion entropy change in each IMF component is obtained to determine the decomposition level *K* through the decomposition of the VDM. Then, the IMF components with useful values are selected to reconstruct the signal, which provides the noise reduction of the vibration signal.

## 3. Time-Frequency Image and Deep Transfer Learning Model

### 3.1. Time-Frequency Image Generation Method

The acquired signal presented by one-dimension, and having non-stationary and nonlinear characteristics, is used as the original vibration signal when data acquisition of the diesel engine is conducted. Nevertheless, three-channel images are used for the input conditions of the data. Therefore, the model is based on the deep learning network to pre-train needs to convert the vibration signal with one dimension into an image with two dimensions. Time-frequency analysis methods are common approaches to extracting fault diagnoses, such as HHT, WT, and STFT [32]. Although the fault characteristic signal can be derived well when the conversion between one-dimension and two-dimension representations is conducted, both WT and HHT methods are relatively slow, in terms of implementation speed.

When considering the conversion speed and including the key feature information concurrently, the STFT method is utilized to produce a time-frequency map represented by two dimensions. Hence, the core idea of the TFT is to add windows to the one-dimensional time-domain vibration signal in segments, and then perform the Fourier transform to run a simultaneous analysis of the time and frequency domain characteristics of the vibration signal, depicted in Figure 1. The STFT is represented by

$$T_{stf,x}(\omega, \tau) = \int_R x(t) w(t - \tau) e^{-j\omega t} dt \tag{8}$$

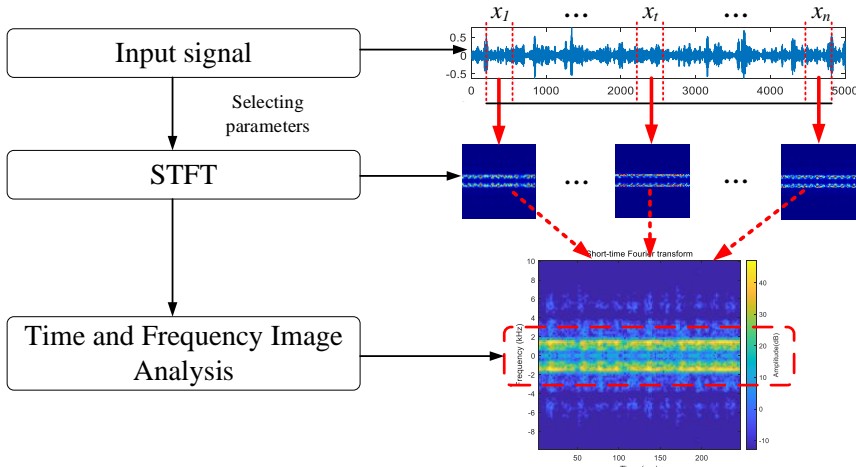

**Figure 1.** The schematic diagram of the basic principle of the STFT.

The discrete form of the STFT is employed to determine the corresponding constituents, and the frequency, amplitude, and phase of each component are given, defined by

$$X[m,k] = \sum_{n=0}^{N-1} x[n]w[n-k]\exp\left(-j\frac{2\pi}{N}mn\right),$$
$$m = 0, 1, \cdots, N-1$$
(9)

where the sequence $x[k]$ represents the sampling signal of the continuous vibration signal $x(t)$; the width of the time window signal $w(t)$ is $N$.

*3.2. Deep Residual Network*

In 2015, the Deep Residual Network (ResNet) model was proposed [33]. When compared with the conventional deep learning network, the ResNet network adds the identity mapping function, which can effectively perform the back-propagation calculation of errors and optimize the hyperparameters of the model. By doing so, the difficulty of network training is considerably decreased. Image processing and recognition methods implement it effectively. Therefore, this manuscript generally deals with the research of the ResNet18 network when diagnosing faults related to machinery.

3.2.1. Convolutional Layer

The neural network (NN) has a core called a convolutional layer that employs a convolution kernel with two dimensions to conduct calculations on the input image. Thus, each pixel in the image is traversed and the feature map is resolved through a nonlinear activation function, depicted in Figure 2, the expression of which is given by

$$x_j^l = \sigma\left(\sum_{i=M_j} x_i^{l-1} \times \omega_{ij}^l + b_j^l\right)$$
(10)

where $l$ denotes the $l$th convolutional layer; $\omega_{ij}^l$ represents the input of the $l$th convolutional layer; $x_j^l$ denotes the output of the $l$th convolutional layer; the bias is denoted by $b_j^l$; $f(\cdot)$ stands for activation function. A Rectified Linear Unit (ReLU) is chosen as the activation function, having expression

$$\text{ReLU}(x) = \max(0, x), x \in (-\infty, +\infty)$$
(11)

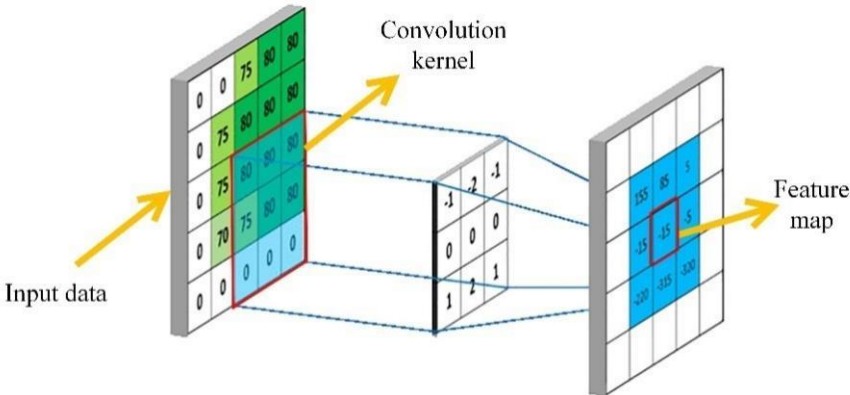

**Figure 2.** The schematic diagram of the convolution process.

### 3.2.2. Max Pooling Layer

The pooling layer comes just after the convolutional layer, and its main purpose is to down-sample the input features by pooling. When the premise of reducing the feature dimension is taken into account, more meaningful features are further extracted. Thus, the parameters of the NN are effectively deceased and the training process would be accelerated. Figure 3 depicts the utilization of the maximum pooling layer. The position-independent characteristic parameters are obtained as the advantage, and its expression is given by

$$x_i^l = \sigma(\beta^l \times down(x_i^{l-1}) + b^l) \tag{12}$$

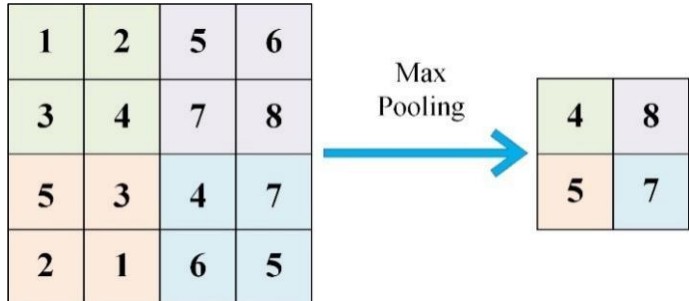

**Figure 3.** Schematic diagram of max pooling operation.

The $down(\cdot)$ represents the pooling function; $b^l$ and $\beta^l$ denote layer l bias and weight, respectively.

### 3.2.3. Residual Building Block

A deep residual network usually consists of multiple residual blocks, which are the basic core of the ResNet network model. The differences between the predicted and the observed values are called residuals. Figure 4 shows two commonly used residual block structures. While Figure 4a shows the standard residual structure block, Figure 4b depicts the residual structure block with a down-sampling layer. The two residual blocks generally consist of two convolutional layers, two batch normalization, and two ReLU activation functions [34]. On the other hand, the residual structure block with the down-sampling layer has a shortcut path directly connecting the input and output, which is practical to conduct backpropagation within the NNs. By doing so, the problem of gradient explosion or disappearance could be resolved effectively, and make network training easier. Therefore, this paper adopts the proposed residual structure block with a down-sampling layer.

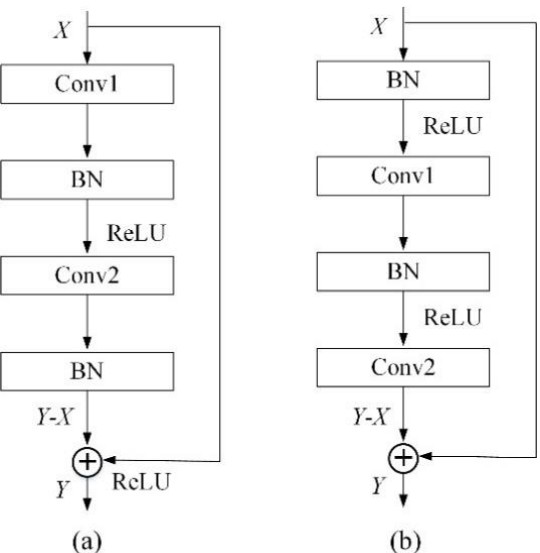

**Figure 4.** The residual structure of the block diagram: (**a**) Indentity residual block; (**b**) Sampling residual block.

3.2.4. Global Average Pooling

To effectively avoid overfitting, a global average pooling layer (GAPL) is used near the output layer of the ResNet network model. A GAPL is utilized to find the mean by utilizing the feature map of each channel and employing it as the output characteristic parameter. Figure 5 depicts this, and Equation (13) presents it.

$$Y_{GAP}(1,1,i_{ch}) = \text{aveage}_{i_{ro},i_{co}}(X_{GAP}(i_{ro},i_{co},i_{ch})) \tag{13}$$

where $Y_{GAP}$ and $X_{GAP}$ represent the output and input of the feature map, respectively.

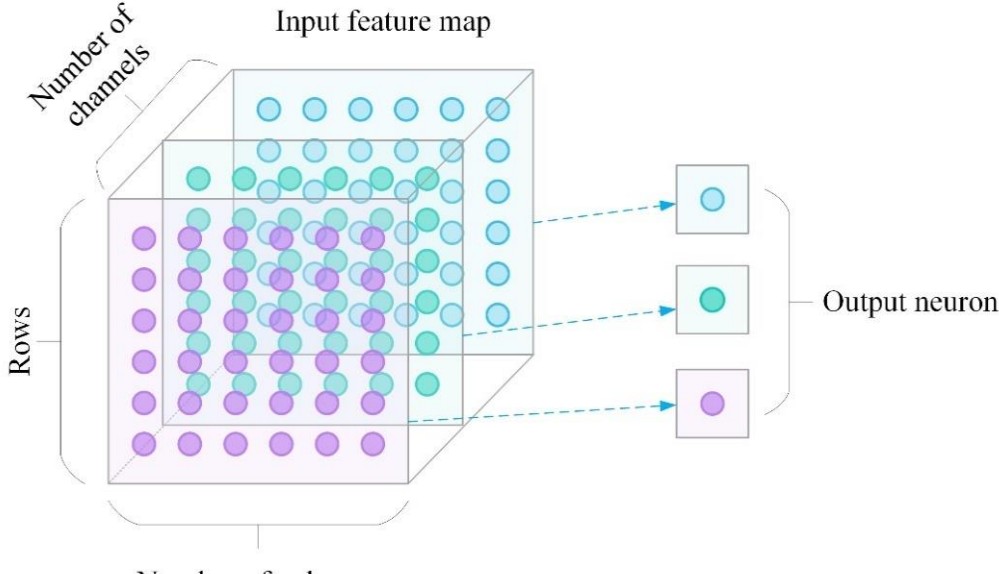

**Figure 5.** The schematic diagram of the global average pooling.

This method could substantially decrease the number of weights necessary for training and could handle translation changes well. In practical applications, the position of the fault vibration signal in the sample is often different, even though the same fault type is observed. The features extracted from the CNNs will also change, and this situation can be summarized as a translational transformation problem. The global average pooling

algorithm can be utilized to resolve the global feature of the average and preserves the invariance of the fault impact location in the deep learning network model. This problem has been improved to some extent, effectively advancing the generalization ability of deep learning network models.

### 3.3. Transfer Learning

Taking data with sufficient numbers of labels from the source domain and transferring the data to a small number of data samples in the target domain is called transfer learning (TL). By running an analysis called sample size analysis, the samples of the source domain are easy to collect and sufficient samples are available [35]. Nevertheless, collecting samples in the target domain is difficult, and relatively few samples are available. When the fault features contained in the trained network model are similar to the new fault features and have potential common data features, employing the transfer method of the model would achieve very ideal outcomes, presented in Figure 6.

$$D = \{W, P(W)\} \tag{14}$$

$$T = \{Y, P(Y/W)\} \tag{15}$$

where $T$ denotes the domain task of the target; $D$ denotes the domain task of the source; $W$ represents the vector space of the features, and $P(W)$ denotes the marginal probability distribution function. This paper adopts a model-based transfer method. From the sampled data of the source and the target domains, the information regarding the model parameter, or the prior knowledge of the model, can be found and shared, which realizes the knowledge transfer.

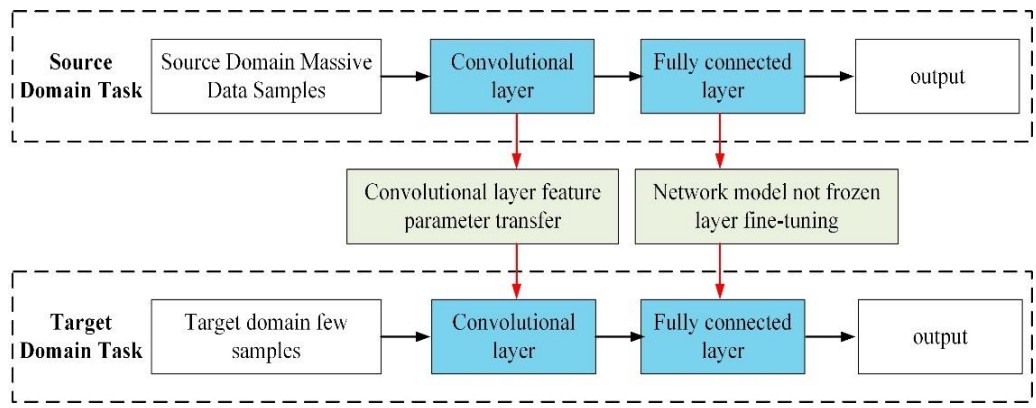

**Figure 6.** The schematic diagram of the model migration process.

## 4. Fault Diagnosis of the Diesel Engine Employing Both Optimized VMD and DTL

Figure 7 depicts the process to diagnose faults utilizing both optimized VMD and DTL, which consists of steps such as data preprocessing, the transformation of a two-dimensional time-frequency map, training network model, and fault classification. More detailed descriptions are given below.

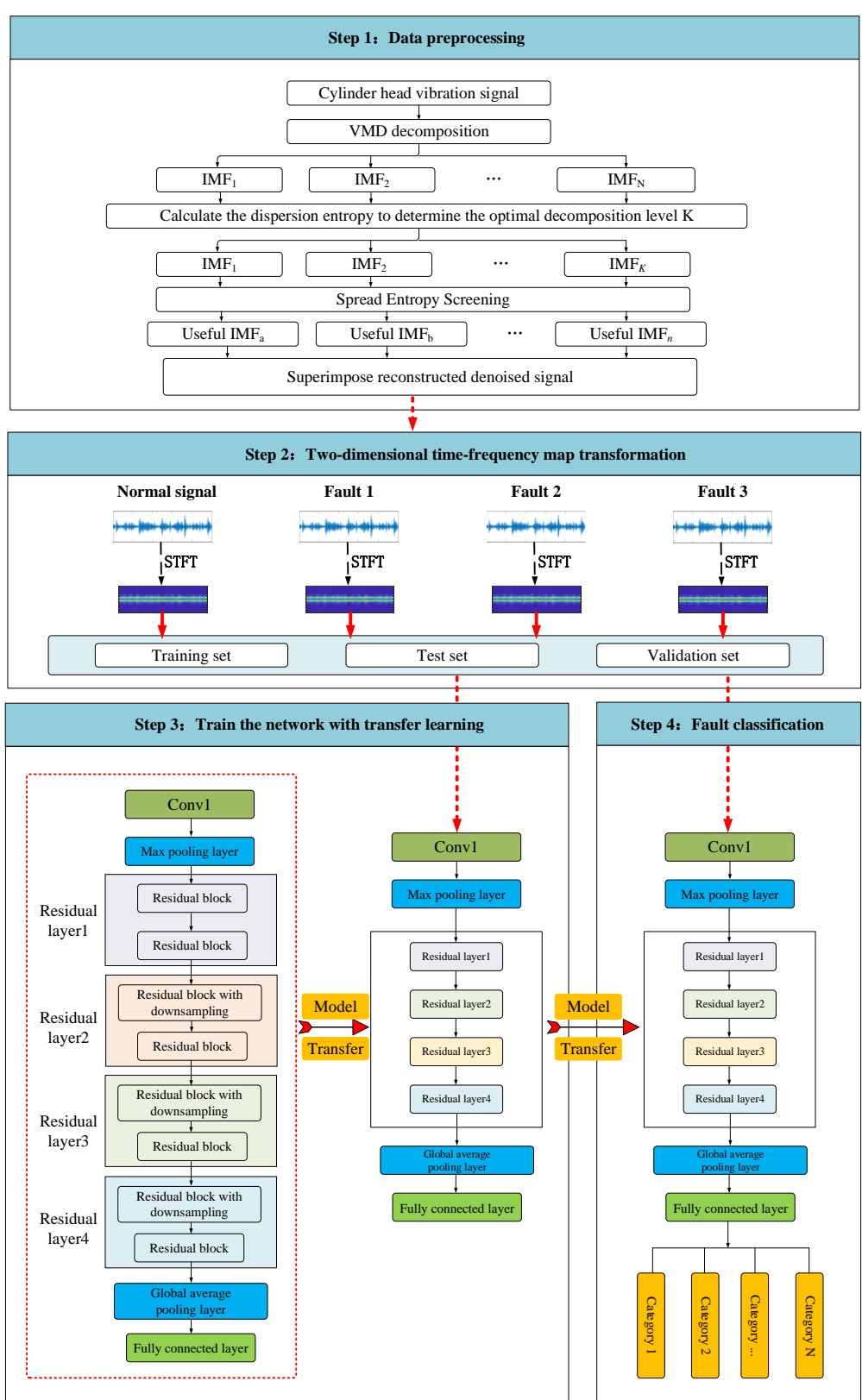

**Figure 7.** The process to diagnose faults of the diesel engine employing optimized VMD and DTL.

**Step 1 Preprocess data.** The VMD algorithm decomposes the original vibration signal, and the scatter entropy value of the IMF component is solved. When the change of dispersion entropy first appears as a turning point, the corresponding decomposition

level is found to be the optimal decomposition level. The valuable information of the IMF component is screened out, according to the dispersive entropy value. Then, it is superimposed and reconstructed to obtain a noise reduction signal.

**Step 2 Transform the two-dimensional time-frequency map.** Taking the noise reduction signal as the input condition, the STFT algorithm is employed to produce the corresponding data set of the two-dimensional time-frequency graph. Then, training, testing, and validation sets are generated, respectively, when the data set is split.

**Step 3 Train the network model.** The parameters of the pre-trained ResNet18 network model are utilized as the transfer object. The parameters of the convolution 1 layer and the residual layer are frozen, and the fully connected layer is fine-tuned. Both training and test data sets are imported into the pre-trained model. The fine-tuned ResNet18 network model is retained to attain a novel DTL-ResNet18 network model.

**Step 4 Classify Fault.** The new test and validation data sets are imported into the trained ResNet18 network model. The Softmax activation function is utilized to obtain the final result of the fault diagnosis.

## 5. The Verification of the Experimental Data

### 5.1. Experiment Preparation

To verify the efficiency of the proposed method to diagnose faults of the diesel engine by employing both optimized VMD and DTL, an in-line 6-cylinder diesel engine test bench was established, which was composed of a panel monitoring condition, and data acquisition of the vibrational signal. A total of 6 vibration sensors were installed to collect data. Figure 8 depicts this.

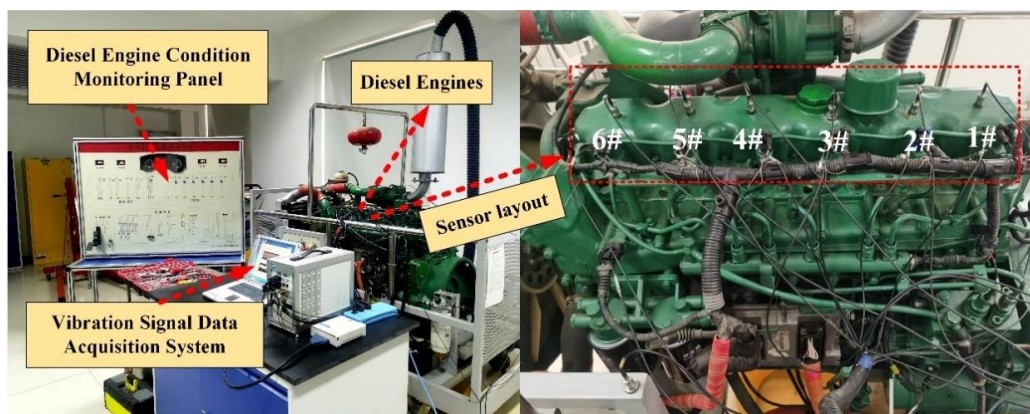

**Figure 8.** The test rig of the planetary gearbox.

During the actual operation of the diesel engine, the probability of failure of the fuel supply system is very high, which not only increases the maintenance cost but also brings great hidden dangers to production safety. Therefore, this paper mainly conducted detailed research on the system of the fuel supply, and preset four failure modes 1–4, as shown in Table 1. The specific three preset faults are shown in Figure 9.

**Table 1.** The pre-set failure modes of the diesel engine.

| Serial Numbers | Fault States | Failure Modes |
| --- | --- | --- |
| 1 | L1 | Normal |
| 2 | L2 | Cylinder misfire |
| 3 | L3 | Air filter clogged |
| 4 | L4 | Broken oil supply pipe |

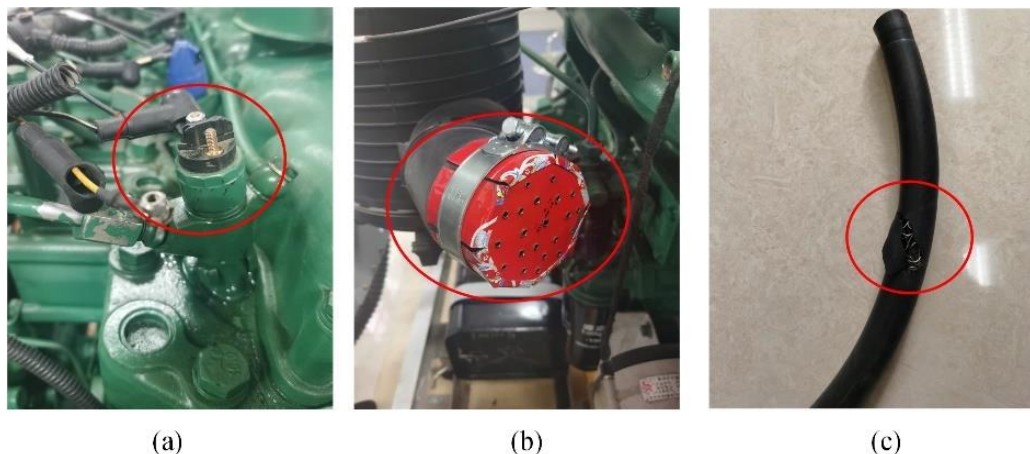

(a)            (b)            (c)

**Figure 9.** The preset failure modes: (**a**) Ring gear, (**b**) Planet gear, (**c**) Sun gear.

### 5.2. Preprocessing and Analysis of the Experimental Data

The presented experiment was to collect effective vibration data. The rotational speed was uniformly set to 800 rpm. Then, 10 groups of data were collected for each failure mode. The frequency of the sampling was assigned to 20 kHz, and the collection time of each group was assigned to 12 s. The next set of data after a 30 s interval was collected. Table 2 presents the detailed data collection parameters of the four preset failure modes. Figure 10 depicts the waveforms of the vibration data of the diesel engine in the four states.

**Table 2.** The datasets of the failures.

| Fault State | Rotating Speed | Sampling Frequency | Sampling Time | Number of Sensors | Number of Samples |
|---|---|---|---|---|---|
| L1 | 800 rpm | 20 kHz | 12 s | 6 | 10 |
| L2 | 800 rpm | 20 kHz | 12 s | 6 | 10 |
| L3 | 800 rpm | 20 kHz | 12 s | 6 | 10 |
| L4 | 800 rpm | 20 kHz | 12 s | 6 | 10 |

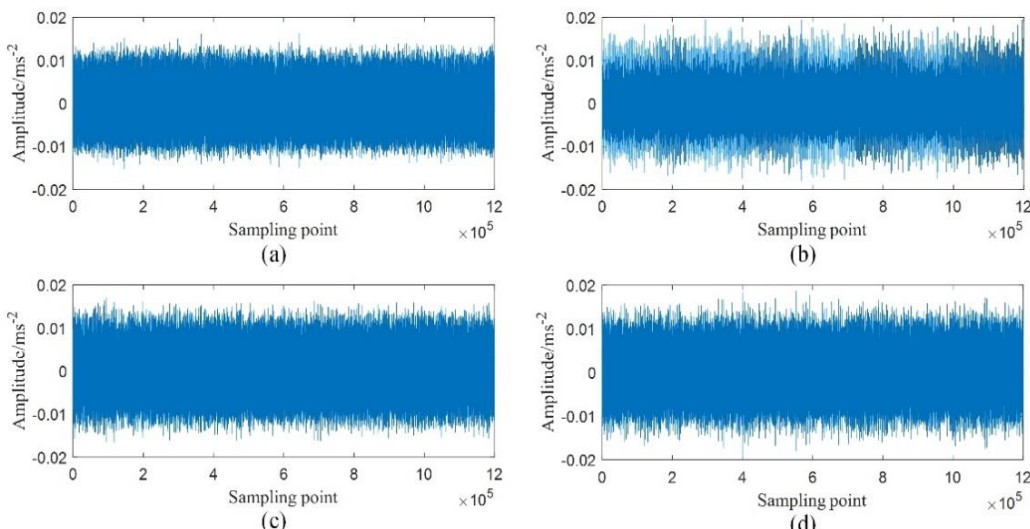

**Figure 10.** The vibration waveforms of the diesel engine for 4 different states: (**a**) L1 state; (**b**) L2 state; (**c**) L3 state; (**d**) L4 state.

When the characteristics of the diesel engine were taken into account, the data of sensor 4 was uniformly used as the verification of this experiment. Due to space issues, only the air filter clogging failure mode was employed as an illustrative example. The

VMD decomposed the original vibration signal. Hence, the dispersion entropy of each IMF component was computed. Therefore, *K*, the end decomposition layer, was determined, and the parameter settings of the spreading entropy were determined. So, the number of categories were denoted by $c = 8$, the dimension by $m = 2$, and the time delay by $d = 1$, respectively. The larger the dispersion entropy of each IMF component, the higher the complexity would be. Besides, more complex interference signals and high-frequency noise signals would be obtained. The dispersion entropy of each component for the VMD decomposition layer was analyzed and the outcomes are presented in Table 3 when various *K* values were utilized. For example, $K = 5$ was the dispersive entropy of each component that had a turning point, indicating the utilization of components and noise components that appeared. Thus, the dispersive entropy of the IMF2 component was found to be the largest one indicating the complexity of the largest signal. Furthermore, the determination of the number of the VMD decomposition layers, which was $K = 5$, was found and the IMF components and spectrograms were eventually decomposed, as shown in Figure 11.

**Table 3.** The analysis of the dispersion entropy for each component of the VMD decomposition.

| IMF | The Dispersion Entropy of Each IMF Component When Various Ks Is Used | | | | |
| | 3 | 4 | 5 | 6 | 7 |
|---|---|---|---|---|---|
| IMF1 | 3.4637 | 3.5874 | 3.7228 | 3.5134 | 3.6991 |
| IMF2 | 3.3146 | 3.4540 | 3.8595 | 3.6764 | 3.7245 |
| IMF3 | 3.0940 | 3.2836 | 3.4521 | 3.6547 | 3.7565 |
| IMF4 | | 3.0683 | 3.2794 | 3.4482 | 3.4512 |
| IMF5 | | | 3.0627 | 3.2768 | 3.2839 |
| IMF6 | | | | 3.0655 | 3.0765 |
| IMF7 | | | | | 2.7068 |

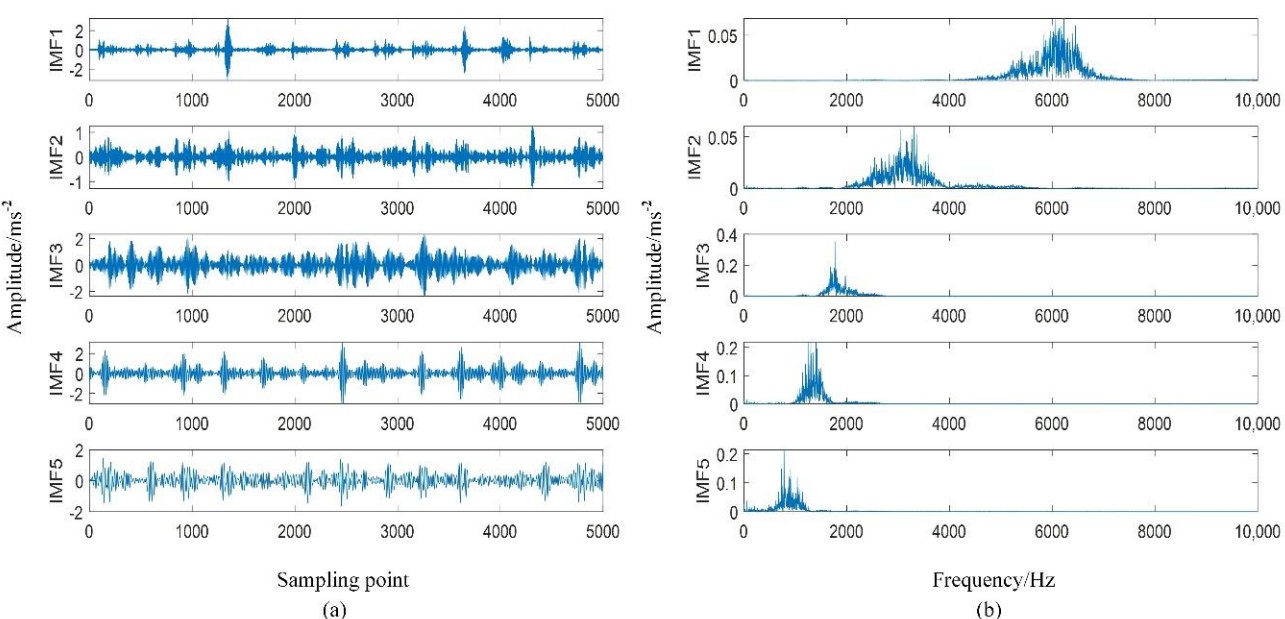

**Figure 11.** The VMD decompositions: (**a**) each IMF component; (**b**) each IMF component spectrogram.

Figure 11b depicts that both IMF1 and IMF2 were high-frequency components with a complex spectrum and wide bandwidths. On the other hand, the IMF3, IMF4, and IMF5 components were selected as the basis for signal reconstruction when both inference signal and complex noise were taken into account. The STFT algorithm was employed to transform the denoised signal after reconstructing the two-dimensional time-frequency map data conducted by the VMD. The sampling length of each fault was uniformly set to 5000, and the size of the time-frequency map was uniformly assigned to $227 \times 227$ pixels.

To meet the requirements of the DTL- ResNet18 network model for the input layer data, the specific two-dimensional time-frequency map data set is presented in Table 4.

**Table 4.** The four preset fault time-frequency graph datasets of the VMD decompositions.

| Signal States | Total Sample | Training Samples | Test Sample | Validation Sample |
|---|---|---|---|---|
| L1 | 240 | 180 | 20 | 40 |
| L2 | 240 | 180 | 20 | 40 |
| L3 | 240 | 180 | 20 | 40 |
| L4 | 240 | 180 | 20 | 40 |
| Total | 960 | 720 | 80 | 160 |

*5.3. The Comparative Analysis of the Noise Reduction Effect*

To further verify the effectiveness of the noise reduction effect of the VMD method, the kurtosis value ($K$) and Peak Signal to Noise Ratio (PSNR) were selected to assess the effect of the noise reduction as indicators, which are defined by

$$K = \frac{\frac{1}{n}\sum_{i=1}^{n}\left(|x_i| - \mu\right)^4}{rms^4} \tag{16}$$

The kurtosis could best represent the shock characteristics of the vibration signal, was more sensitive to the vibration shock, and was around 3 in the case of passing. If the deviation would be too large, it meant that the equipment was subjected to a certain vibration and shock, or there could a hidden danger of failure related to a key component.

$$PSNR = 101g(z_{max}^2 / (\frac{1}{N}\sum_{j=1}^{N}\left(z_j - z'_j\right)^2)) \tag{17}$$

where $Z$ and $Z'$ denote the original and constructed vibration signals, respectively. *PSNR* is an important indicator directly reflecting the capacity of the noise reduction for the vibration signal. When *PSNR* became higher, a better noise reduction effect could be observed.

Table 2 presents the L4 fault data that was uniformly used to conduct validity. It can be seen from Table 5 that, after decomposing the vibration signal with strong noise by the VMD method, the kurtosis became the highest, indicating that the equipment deviated from the normal state, which was a more obvious characteristic of the fault. Thus, this suggested that the VMD method could eliminate strong noise signals and had a strong ability to retain fault features. When the *PSNR* index was compared with the CEEMD, EEMD, and EMD methods, the *PSNR* value of the VMD method reached up to 32.983. Hence, the higher the *PSNR* value, the better the noise reduction effect would be. Therefore, it was verified that the VMD method not only had a better noise reduction impact, but also reduced the difficulty of extracting fault features. Then, a basis to improve the accuracy of the fault diagnosis was provided in the next step.

**Table 5.** The contrastive analysis of the noise reduction with different modal decompositions.

| Evaluation Indicators | Vibration Signal | CEEMD | EEMD | EMD | VMD |
|---|---|---|---|---|---|
| $K$-value | 3.362 | 3.747 | 4.253 | 4.641 | 5.385 |
| PSNR | 4.253 | 15.652 | 18.368 | 24.527 | 32.983 |

*5.4. The Setup and Training of the Network Model*

The quality of the construction of the DTL-ResNet18 network model directly affected the final results of the fault classification. The detailed parameter settings for the training of the network model are shown in Table 6. To verify the effectiveness of the training of the network model, the data set presented in Table 4 was utilized for verification. The t-distributed Stochastic Neighbor Embedding (t-SNE) method was employed to extract the

image features of the pre-trained ResNet18 network model [36]. Then, the analysis of the data visualization was performed. Figure 12 depicts this.

**Table 6.** Setting the model parameters of the ResNet18 network.

| Network Layer | Type | Size × Number of Output Channels | Output (Size × Number of Channels) |
|---|---|---|---|
| Input layer | RGB | | $227 \times 227 \times 3$ |
| Conv1 | Convolutional Layer | $7 \times 7 \times 54$ | $114 \times 114 \times 64$ |
| Maxpool | Max Pooling Layer | $3 \times 3 \times 64$ | $57 \times 57 \times 64$ |
| Layer1 | Residual Layer: IRB64 + IRB64 | $3 \times 3 \times 64$<br>$3 \times 3 \times 64$<br>$3 \times 3 \times 64$<br>$3 \times 3 \times 64$ | $57 \times 57 \times 64$<br>$57 \times 57 \times 64$<br>$57 \times 57 \times 64$<br>$57 \times 57 \times 64$ |
| Layer2 | Residual Layer: SRB128 + SRB128 | $3 \times 3 \times 128$<br>$3 \times 3 \times 128$<br>$1 \times 1 \times 128$<br>$3 \times 3 \times 128$<br>$3 \times 3 \times 128$ | $29 \times 29 \times 128$<br>$29 \times 29 \times 128$<br>$29 \times 29 \times 128$<br>$29 \times 29 \times 128$<br>$29 \times 29 \times 128$ |
| Layer3 | Residual Layer: SRB256 + SRB256 | $3 \times 3 \times 256$<br>$3 \times 3 \times 256$<br>$1 \times 1 \times 256$<br>$3 \times 3 \times 256$<br>$3 \times 3 \times 256$ | $15 \times 15 \times 256$<br>$15 \times 15 \times 256$<br>$15 \times 15 \times 256$<br>$15 \times 15 \times 256$<br>$15 \times 15 \times 256$ |
| Layer4 | Residual Layer: SRB512 + SRB512 | $3 \times 3 \times 512$<br>$3 \times 3 \times 512$<br>$1 \times 1 \times 512$<br>$3 \times 3 \times 512$<br>$3 \times 3 \times 512$ | $8 \times 8 \times 512$<br>$8 \times 8 \times 512$<br>$8 \times 8 \times 512$<br>$8 \times 8 \times 512$<br>$8 \times 8 \times 512$ |
| Avapool | Global average pooling layer | $8 \times 8 \times 512$ | $1 \times 1 \times 512$ |
| FC | Fully connected layer | $1 \times 1 \times 4$ | 4 |

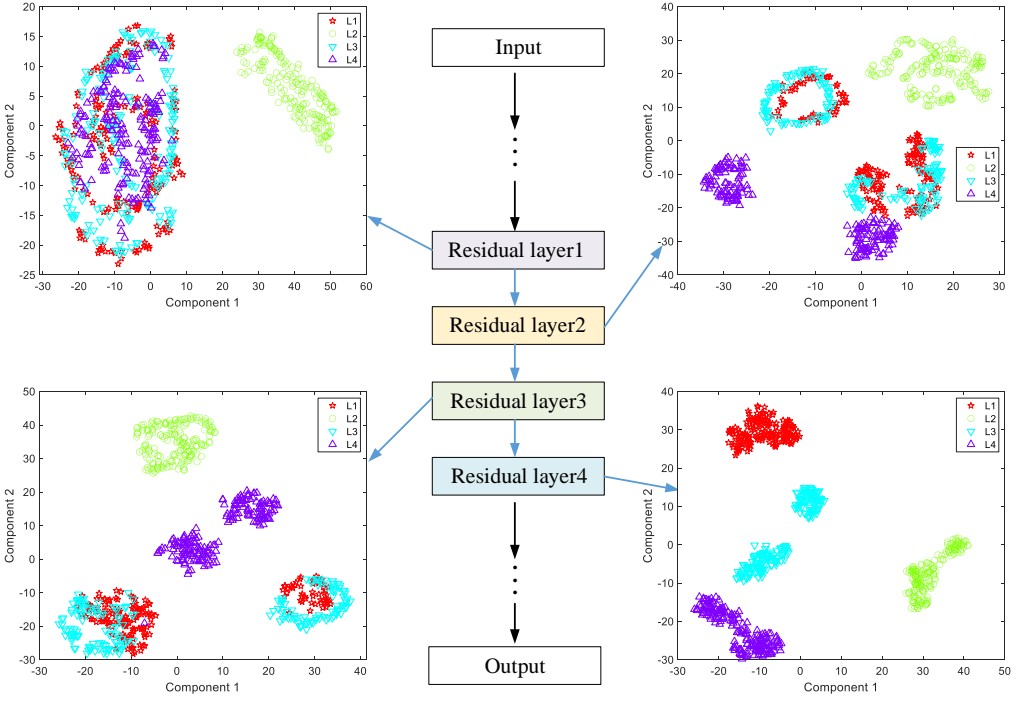

**Figure 12.** The t−SNE visualization analysis of the VMD−ResNet18 network for feature extraction.

Figure 12 depicts that extracting features related to faults in residual layer 1 could only distinguish the fault feature parameters of L2. On the other hand, the other three types of faults overlapped, thus, segregation became difficult. Then, the fault features extracted in the residual layer 2 could effectively distinguish the fault features of both L2 and L4. However, there still existed some fault features that had not been distinguished. When the fault features were extracted by residual layer 3, the fault features of both L1 and L3 also crossed each other. Eventually, the fault features extracted in the residual layer 4 could clearly distinguish four fault features, and the clustering effect was found to be good. Therefore, this could effectively suggest the effectiveness of the pre-trained ResNet18 network model to determine features for fault diagnosis with a strong ability to extract features. Table 7 presents the better fault classification and processing capabilities of the pre-trained network model and its application status for the transfer of network models.

**Table 7.** The pre-training diagnostic results of the VMD-ResNet18.

| Network Model | Diagnostic Results | | | | Training Time/S |
|---|---|---|---|---|---|
| | L1 | L2 | L3 | L4 | |
| VMD-ResNet18 | 100.0% | 100.0% | 100.0% | 100.0% | 121.635 s |

*5.5. The Contrastive Analysis of the Decomposition Diagnosis Results with Various Modes*

To further verify the pre-trained ResNet18 network model and the effectiveness of the optimized VMD method suggested in the manuscript, they were compared with the CEEMD, EEMD, and EMD algorithms, respectively. First, the original vibration signals were denoised by the VMD, CEEMD, EEMD, and EMD methods, respectively. Secondly, the reconstructed noise-reduced signal was transformed into a two-dimensional time-frequency graph by the STFT algorithm. The data set of each fault contained 180 training samples and 20 test samples, respectively. Finally, the pre-trained ResNet18 network model was uniformly employed to diagnose faults. Table 8 presents the outcomes.

**Table 8.** The results of the contrastive analysis of the various mode decomposition diagnosis.

| Fault State | Different Modal Decomposition Methods | | | |
|---|---|---|---|---|
| | CEEMD | EEMD | EMD | VMD |
| L1 | 97.92% | 100.0% | 100.0% | 100.0% |
| L2 | 100.0% | 100.0% | 100.0% | 100.0% |
| L3 | 87.50% | 95.83% | 93.75% | 100.0% |
| L4 | 93.75% | 87.50% | 95.83% | 100.0% |
| Accuracy | 94.79% | 95.83% | 97.40% | 100.0% |
| Training time/s | 166.744 s | 137.427 s | 153.226 s | 121.635 s |

The diagnostic results suggested that the diagnostic accuracy of the VMD method reached 100%, and the lowest diagnostic result of the CEEMD was only 94.79% when compared to the three approaches. When the network training time was under consideration, the VMD algorithm had less training time than the others. Besides, the accuracy was higher, as shown in Figure 13. The verification results suggested that the proposed method could effectively remove both interference signals and complex noises. Hence, the extracted and reconstructed signal contained the required fault features, so that the pre-trained ResNet18 network model could obtain higher diagnostic accuracy.

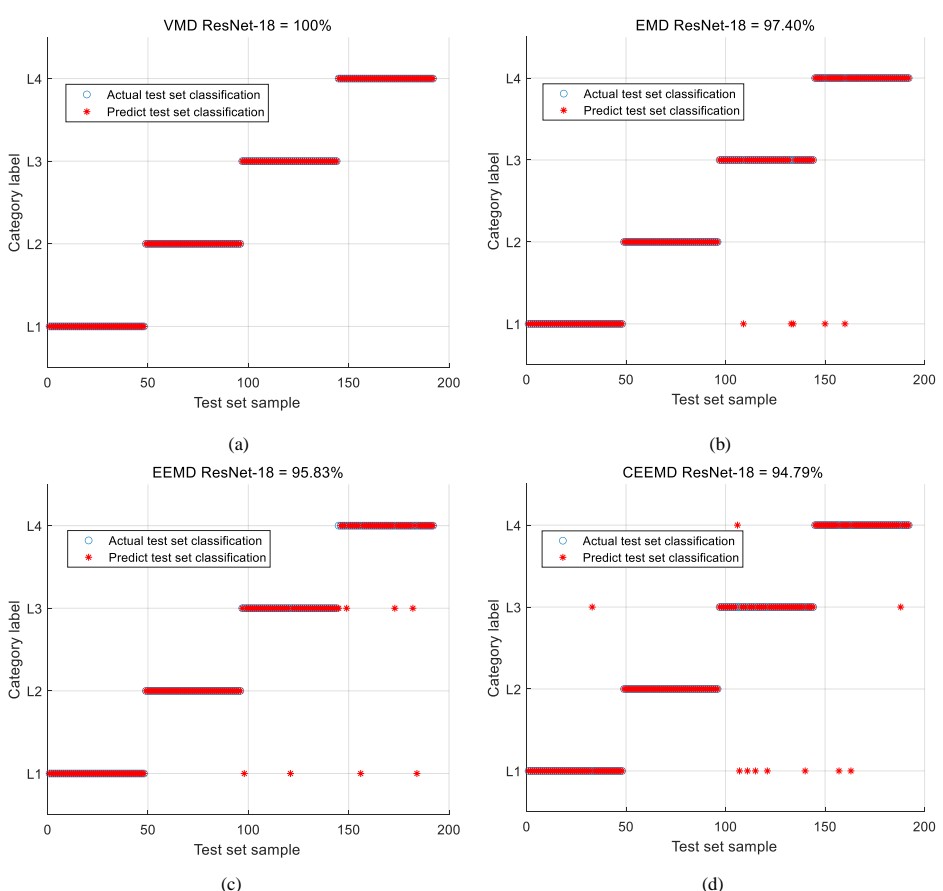

**Figure 13.** The results of the contrastive analysis of the different mode decomposition diagnoses:
(**a**) VMD = 100%; (**b**) EMD = 97.4%; (**c**) EEMD = 95.83%; (**d**) CEEMD = 94.79%.

*5.6. Comparison and Analysis of Diagnosis Results of Different Network Models*

To validate the effectiveness of the ResNet-18 network model, based on DTL proposed in this paper, AlexNet, SqueezeNet, and GoogLeNet network models were employed to make comparisons. The data set in Table 4 was used. The main network parameter settings of the four network models are shown in Table 9.

**Table 9.** Comparing parameters of the four methods.

|  | **ResNet18** | **AlexNet** | **SqueezeNet** | **GoogLeNet** |
| --- | --- | --- | --- | --- |
| Solver Name (Algorithm) | sgdm | sgdm | sgdm | sgdm |
| Initial Learn Rate | 0.00005 | 0.00005 | 0.00005 | 0.00005 |
| Max Epochs | 10 | 10 | 10 | 10 |
| Mini Batch Size | 15 | 15 | 15 | 15 |
| Validation Frequency | 3 | 3 | 3 | 3 |
| Shuffle | Every epoch | Every epoch | Every epoch | Every epoch |
| Execution Environment | GPU | GPU | GPU | GPU |

From Table 10 it can be seen that the accuracy rate of the ResNet18 network model reached 94.27% when the original data was directly used for 2D time-frequency map conversion. The accuracy ratio was higher when the comparison was made with the three approaches, which verified the effectiveness of the network model. After preprocessing of the IMF component of the original data by the VMD, the accuracy of the ResNet18 network model reached 100.0%, which was the highest when different modal decompositions were under consideration. The experimental results suggested that the VMD method had a better preprocessing ability when dealing with complex and strong noise signals. Besides, not only could the interference signal be effectively eliminated, but also higher diagnostic

accuracy could be obtained. When both different network models and the unified network parameter settings were a concern, the ResNet18 network model had the best diagnostic results. However, the diagnostic results of the GoogLeNet network model did not achieve the same diagnosis effect. Therefore, both the effectiveness and feasibility of the ResNet18 network model once more were attained.

**Table 10.** The comparison of the fault diagnosis of the different network models.

| Network Model | Accuracy | | | | |
|---|---|---|---|---|---|
| | RAW DATA | CEEMD | EEMD | EMD | VMD |
| ResNet18 | 94.27% | 94.79% | 95.83% | 97.39% | 100.0% |
| AlexNet | 88.02% | 69.27% | 79.16% | 78.13% | 95.31% |
| SqueezeNet | 61.97% | 65.10% | 60.41% | 71.35% | 83.85% |
| GoogLeNet | 62.50% | 56.77% | 41.14% | 48.44% | 76.56% |

Both GoogLeNet and AlexNet network models showed stronger feature extraction and network learning capabilities when both classification and recognition were a concern. However, utilizing them for fault diagnosis cannot generate better outcomes when complex nonlinear vibration signals are under consideration, as, for example, with diesel engines. Figure 14 depicts that the method combining VMD and ResNet18 network model had better noise reduction performance and better diagnosis effect. The proposed method helped find a more efficient way of diagnosing faults of the diesel engine.

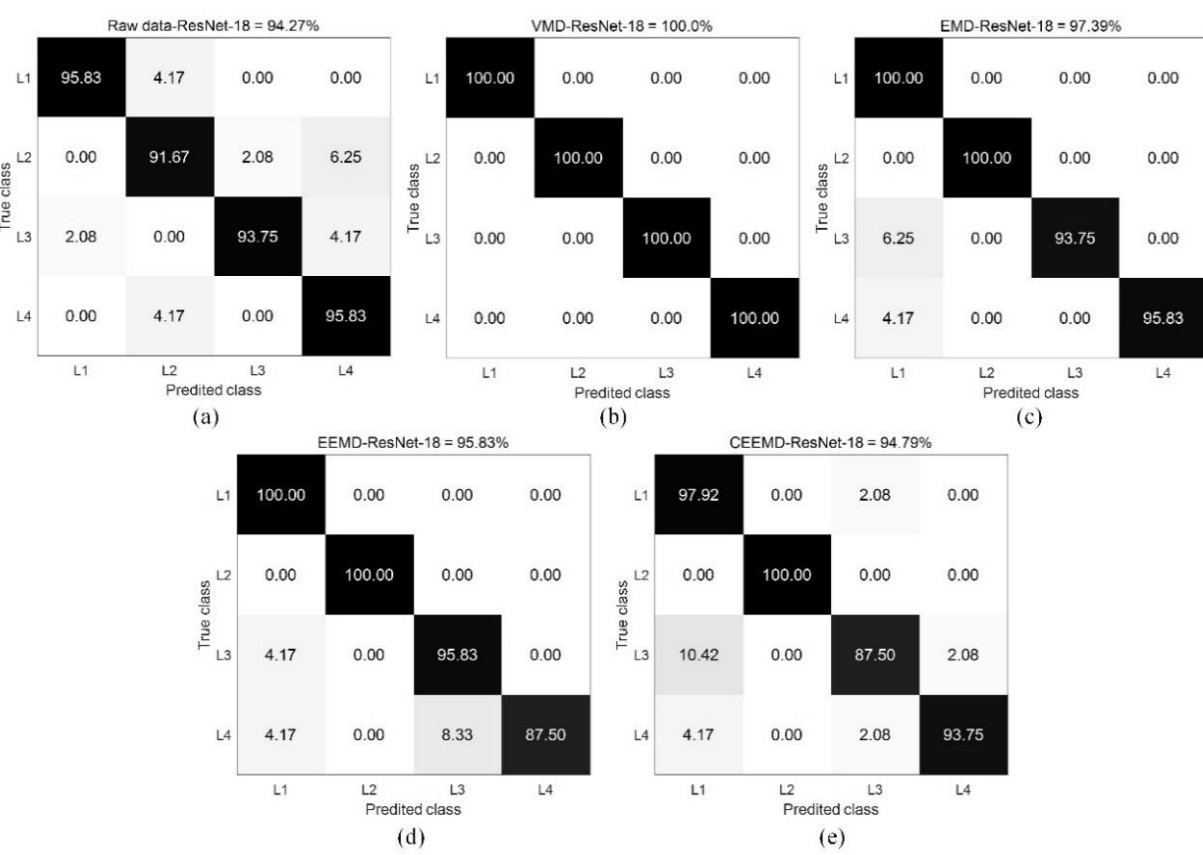

**Figure 14.** The fault diagnosis results of the different network models: (**a**) Raw data = 94.27%; (**b**) VMD-ResNet18 = 100%; (**c**) EMD-ResNet18 = 97.39%; (**d**) EED-ResNet18 = 95.83%; (**e**) CEED-ResNet18 = 94.79%.

## 6. Conclusions

This research suggested an approach to diagnosing faults utilizing the combination of optimized VMD and DTL, concurrently. A successful transference of the pre-trained ResNet18 model on the ImageNet samples was conducted to diagnose faults of diesel engines in a more complex noise environment. The manuscript contributes the following:

(1) To decompose the level selection of the VMD, the level selection method of dispersive entropy was adopted and better noise reduction impact was attained.

(2) A two-dimensional time-frequency image processing problem with noise reduction was dealt with, after employing the STFT method to convert the one-dimensional vibration signal so as to extract features.

(3) An approach dependent upon the DTL was suggested. The ResNet-18 network was selected as the transfer object, and the network was fine-tuned. The proposed approach could directly derive key features of faults from two-dimensional time-frequency images and perform fault diagnosis. Thus, the difficulty of manually extracting features and the dependence on expert experience were reduced. Therefore, both diagnostic efficiency and accuracy were greatly improved.

The experiments showed that the implementation impact of the DTL concerning the diagnosis of faults for mechanical parts was better than other network models. As artificial intelligence technology advances rapidly, the DTL would play a key role in various engineering applications. Besides, the results of this research will help expand new knowledge in this field and have good reference value.

**Author Contributions:** Data curation, H.B. and X.Z.; Resources, L.W. and H.Y.; Supervision, X.J.; Validation, L.W. and X.J.; Writing—original draft, H.B.; Writing—review & editing, H.B. All authors have read and agreed to the published version of the manuscript.

**Funding:** This study did not receive any funding.

**Conflicts of Interest:** The authors declare that they have no conflict of interest.

## Abbreviations

| Symbols | Full Explanations |
| --- | --- |
| VMD | Variational Mode Decomposition |
| IMF | Intrinsic Mode Function |
| HHT | Hilbert Huang Transform |
| WT | Wavelet Transform |
| STFT | Short Time Fourier Transform |
| DE | Dispersion Entropy |
| IMF | Intrinsic Mode Function |
| EMD | Empirical Mode Decomposition |
| EEMD | Ensemble Empirical Mode Decomposition |
| CEEMD | Complete Ensemble Empirical Mode Decomposition |
| LMD | Local Mean Decomposition |
| ADMM | Alternate Direction Method of Multipliers |
| PSNR | Peak Signal to Noise Ratio |
| ReLU | Rectified Linear Unit |
| t-SNE | t-distributed Stochastic Neighbor Embedding |
| ResNet | Residual Network |

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
