# Peer review of "Combination of Optimized Variational Mode Decomposition and Deep Transfer Learning: A Better Fault Diagnosis Approach for Diesel Engines"

_electronics, doi:10.3390/electronics11131969_

Round 1
Reviewer 1 Report
The paper looks nicely presented, however upon reading it, many issues between major and minor appears as follows:
1. You need to justify the novelty of the contributions that are mentioned in Sec1.
2. What is the motivation? This seemed a bit vague.
3. Some equations are big, but shortly explain. Such as equation 3 and 4. Can you please elaborate further.
4. I cannot get to understand the figures, such as figure 7. Why 3 parts?
5. Results are adequate, but in figure 12 plots, where are the units you used to measure these features.
6. You need to add a table of those notations you used in the paper.
7. Proofread the paper thoroughly.
Author Response
comments to the responses are attached

Reviewer 2 Report
General Comments:
Literature and background foundations are well drawn.
Line 303: It is stated “in order to ensure the stability of the collected data”, this poses serious limitations to this study since uniform speed is seldomly attained in real operation! Should be addressed or otherwise referred as a limitation.
Line 390: It is stated the VMD denoises acquired data which is not necessarily true! VMD decomposes the signal into several components, as demonstrated into important ones towards monitoring, but it does not “denoise”! Again in Line 392 it is stated, “It is fully proved that the VMD algorithm can retain valuable fault signal data after effectively 392 eliminating strong noise signals.” Which is not proved! Rephrase or eliminated!
The methods/methodology seems robust, and results appear to attain significant improvements, but some allegations should be made with care! Further, the paper provides a new approach but lacks reasoning behind these improvements. Nevertheless, it is sound and contributes to new knowledge on the field.
Structure and Content:
Author filiation is not accurate, missing numbering.
The abstract requires minor English revision.
Line 38: Alter capital A. Further Capital letters appear after commas!
Lines 39-40: Rephrase.
Line 100: Irregular start of phrase that should be revised. It should not start with abbreviations.
Lines 204-204: Phrase should be rephrased.
Lines 190-246: The text explains in some detail the background of deep belief networks, but this can be found easily in literature. Since no derivations or otherwise considerations are lately taken upon this subject it should be significantly reduced.
Author Response
Responses to the comments are attached

Round 2
Reviewer 1 Report
The authors addressed all my comments and the amount of work they have done is clear in the revised version.